# Yield Components and Development in Indeterminate Tomato Landraces: An Agromorphological Approach to Promoting Their Utilization

Adolfo Donoso [1] and Erika Salazar [2,*]

1  Horticultural Crops, Instituto de Investigaciones Agropecuarias, Santiago 8831314, CP, Chile
2  Genetic Resources Unit and Germplasm Bank, Instituto de Investigaciones Agropecuarias, Santiago 8831314, CP, Chile
*  Correspondence: esalazar@inia.cl

**Abstract:** Nowadays, increments in tomato yield seem to have reached a plateau. Tomato genebank collections have been recognized as a novel source for yield increments. The use of the diversity in Latin America for novel improved varieties is limited by the knowledge gap regarding field-grown tomatoes. As yield has complex, unresolved trade-offs, agromorphological traits become useful for further improvement. In this study, the development of successive clusters was studied in twenty-four Chilean tomato landraces to elucidate the relationships among agromorphological traits of flowers, inflorescences, and fruits. Plants yielded an average of 3297 g m$^{-2}$, with a variation coefficient of 0.44. Correlation analyses were performed to evaluate the relationships between yield components and plant phenology. Findings suggested a two-level compensation between average fresh fruit weight and the number of fruits, one on a plant basis and the second on a cluster basis. All traits evaluated had significant phenotypic correlations with yield traits. Growing degree days for a cluster to develop had a low negative phenotypic correlation with yield (−0.33***) and a high genetic correlation with the number of clusters (−0.90***). The number of set flowers, as opposed to the number of flowers, was significantly correlated with average fresh fruit weight (−0.17***), supporting the initiation of the trade-off after the fruit set. This study provides new insight into the plant agromorphology of indeterminate plants. In a global climate change context, further study of trade-off relationships is important for identifying genotypes able to sustain their productivity.

**Keywords:** *Solanum lycopersicum*; indeterminate plant agromorphology; plant development; yield components; tomato landraces; Chile





## 1. Introduction

The tomato (*Solanum lycopersicum* L.) is a major horticultural crop worldwide. After significant increments in the yield of tomato cultivars over the past 50 years, nowadays the increments in yield seem to have reached a plateau [1]. Tomato genebank collections have been recognized as a source of diversity that could contribute to yield increments [2]. Multiple studies have described the yield and development of commercial cultivars [1,3–6], the genetic regulation associated with fruit-morphology diversity [7–9], and the phenotypic distinctions between landraces [10–12]. Nevertheless, the knowledge of field-grown tomatoes is behind the understanding of other crops [13]; there are few studies evaluating the diversity of tomato landraces regarding yield [14], physiological yield component compensation [15], or yield-associated traits [16].

The recognition of trade-offs is a main component for further improvement [17]. This is needed today in order to develop novel improved varieties that sustain their productivity under abiotic stresses imposed by climate change [18]. Yield components are not independent, but are integrated at different levels. Interaction effects make yield a low-heritability trait, therefore, for the comparison of varieties, physiological and agromorphological traits

become useful. Agromorphological traits are stable and tend to be of high heritability [19]. In tomatoes, larger flowers [20,21], inflorescence traits, and fruit-morphology traits [22] have been associated with fresh fruit weight. However, under water deficit, an increase in fruit weight occurs due to the decrease in the number of flowers [23], while compensation between tomato clusters has been established, in both the fresh weight of fruits and fruit number, related to competition between clusters in a source-sink balance [7,24–26]. This has been recently reviewed by Nesbitt [7], where the QTL fw2.2 was shown to be responsible for 30% of the difference in fruit weight in tomatoes, due to a modification of the source-sink relationship. However, this modification in fruit weight does not affect the total yield or harvest index under non-stress conditions.

The potential yield increment of crops has been associated with a crop morphophysiological approach and broadening the genetic base [27]. As with other cultigens, tomatoes had a two-stage path of dispersal; one in the pre-Columbian period in America with a subsequent introduction to Europe before the 1540s [28], and a modern post-Columbian reintroduction, where a narrow genetic base of tomato varieties displaced traditional tomato landraces [29]. In the 1950s, Rick [30] reported enormous variation in the tomato landraces of Ecuador, Peru, and Chile, in the Western Andes, and high segregation in mid-twentieth-century tomato collections from Latin America. Nowadays, the diverse Latin American tomato landraces have the potential to be the precursors of improved varieties, but the use of this diversity is limited due to the incomplete descriptions of their agromorphological, phenotypic, genetic, and biochemical traits [2]. Thus, the objective of this work was to evaluate the fresh yield components of tomatoes, in diverse tomato landraces from a Chilean collection of tomatoes, under on-field cultivation; this was achieved by studying the development of successive tomato clusters, in order to elucidate the relationships between the agromorphological traits of flowers, inflorescences, and fruits.

## 2. Materials and Methods

### 2.1. Plant Material

A total of 24 accessions of tomato (*Solanum lycopersicum* L.) were characterized. The accessions were selected to represent a broad diversity of fruit morphologies, origins, and years of collection (Table 1). Almost half of the accessions correspond to Limachino types, though not all Limachino tomatoes are the same, as they have been collected from different growers during the past century.

### 2.2. Essay

An on-field essay was carried out in the 2016–2017 season at La Platina Regional Research Center, Instituto de Investigaciones Agropecuarias, located at the Región Metropolitana de Santiago, Chile (33°34′20.20″ S, 70°37′31.22″ O, 620 m.a.s.l). The environmental conditions during the evaluation season are detailed in Table S1. The essay was designed as a randomized complete block with subsampling, with 3 blocks and a subsampling of 6 plants per block for a total of 432 plants. Seedlings were sown on 31 August and transplanted on 7 October, with a spacing of 0.5 m between plants and 0.7 m between rows. The essay was uprooted regularly during the season, in order to maintain the plants in a single axis. Staking was performed with one colihue stake for each stem. Mulch was used to control weeds. A weekly fertirrigation was applied in the same manner as that used commercially, with 0.75 kg of N and 1.3 kg of $K_2O$ initially, and after the second cluster, 0.52 kg of N and 1.3 kg of $K_2O$ every week. Irrigation was sub-optimal, with 2 irrigations missed due to technical issues, both occurring during the harvest period. On December 22nd, a preventive application of Karate Zeon (200 cc ha $^{-1}$) and Neres 50% SP (1.0 kg ha $^{-1}$) was applied, for aphids (*Aulacorthum solani*, *Aphis craccivora*, *Aphis gossypii*, *Myzus persicae*) and tomato moths (*Tuta absoluta*), respectively.

The measurements were taken throughout the season on the third cluster of each plant, according to the independent phenology. Flower measurements were taken very early in the season, taking one flower by a destructive method for indoor evaluation.

**Table 1.** Information on the tomato landraces used in this study.

| Collection ID [1] | Other ID [2] | Name | Collection Site and Year | Description |
|---|---|---|---|---|
| SLY49 | PI128586 | - | Limache, Chile 1938 | Small, cherry-like |
| SLY30 | PI128587 | - | Limache, Chile 1938 | Flat, highly ribbed |
| SLY50 | PI128588 | - | Limache, Chile 1938 | - |
| SLY74 | PI264548 | Limachino | Campex Los Andes, Chile 1960 | High locule number, early |
| SLY148 | | Limachino | Limache, Chile 1980 | High locule number, early |
| SLY149 | | Limachino | Limache, Chile 1980 | High locule number, early |
| SLY150 | | Limachino | Limache, Chile 1980 | High locule number, early |
| SLY151 | | Limachino | Limache, Chile 1980 | High locule number, early |
| SLY152 | | Limachino | Limache, Chile 1980 | High locule number, early |
| SLY147 | | Limachino | Limache, Chile 1980 | High locule number, early |
| SLY121 | | Limachino Español | Limache, Chile 2015 | High locule number, early |
| SLY122 | | Limachino Frances | Limache 2015 | High locule number, early |
| SLY123 | | Limachino Italiano | Limache 2015 | Beefsteak, pear-shaped |
| SLY124 | | Limachino | Limache 2015 | High locule number, early |
| SLY82 | PI270198 | Marglobe | USA 1960 | Round, smooth |
| SLY83 | PI157850, CGN14430 [3] | Marmande | Israel 1947 | High locule number |
| SLY129 | | Marmande | Chile, 1980 | High locule number |
| SLY65 | PI128611 | - | Antofagasta, Chile 1938 | Lanceolate leaf, round, pink |
| SLY66 | PI128612 | - | Antofagasta, Chile 1938 | Small, round |
| SLY70 | PI128618 | - | Tacna, Perú 1938 | Flat, small to medium |
| SLY39 | PI128615 | - | Arica, Chile 1938 | Lanceolate leaf, variable fruit |
| SLY47 | PI128447 | - | Talca, Chile 1938 | - |
| SLY159 | | Rosado | San Clemente, Chile 2015 | Beefsteak, pink |
| SLY157 | | Rosado | Coihueco, Chile 2015 | Beefsteak, pink |

[1] http://www.recursosgeneticos.com/ accessed on 13 December 2021, [2] https://www.ars-grin.gov/ accessed on 13 December 2021, [3] https://cgngenis.wur.nl/ accessed on 13 December 2021.

*2.3. Harvest*

A continuous harvest was carried out during the season, ending 144 days after transplanting on 28 February. With the time interval between harvests of the same plant not exceeding 10 days, a total of 16 harvest events were made during the season. At harvest, each plant and cluster were identified independently, and the number of fruits (cluster $^{-1}$, plant $^{-1}$, time of harvest $^{-1}$) and the fresh weight of the fruits (g cluster $^{-1}$, plant $^{-1}$, time of harvest $^{-1}$) was determined.

*2.4. Yield, Yield Components, and Phenotypical Traits*

Trait measurements (Table 2) were taken for each plant following IPGRI guidelines at each plant's third-cluster phenological stage. Flower measurements were taken very early in the season, taking one normal flower by a destructive method for indoor evaluation. The style length (STL) was used along with the ovary length (OL) and anther length (AL) to estimate the stigma exertion (STE):

$$STE = STL + OL - AL \tag{1}$$

At the end of the season, the plant traits were estimated and expressed as m $^{-2}$ to account for plant density: total fresh fruit yield (Y) was determined by calculating the sum of the harvests made on each plant during the season, and expressed as g m $^{-2}$; the total number of fruits harvested per plant (NFM) was determined by calculating the sum of the fruits counted in each harvest of each plant, independently of the cluster of the plant, and expressed as fruits m $^{-2}$; average fresh fruit weight (FFW) was determined by calculating the total fresh fruit yield (Y) divided by the total number of fruits (NFM), and expressed as g per fruit; the number of clusters per plant (CLU) corresponded to the count of the clusters that yielded a red, harvestable tomato, expressed as clusters m $^{-2}$; the average number of

fruits harvested per cluster (FRU) was determined by dividing the total number of fruits (NFM) by the number of clusters (CLU). The growing degree days were calculated with a base temperature of 10 °C [14]. The date of harvest of the first fruit of each cluster of each plant was recorded and used to determine the growing degree days to the harvest. The growing degree days to the first tomato harvested on the last cluster of each plant were divided by the number of clusters of the plant (CLU), to estimate the mean thermal time for a cluster to develop (CGDD). To assess in more detail the within-cluster compensation of each cluster of each plant, the tomatoes harvested from each cluster in each plant were counted, weighed and the harvest date was registered, to determine the following: the total number of fruits harvested from a single cluster (NFC); the total fresh fruit weight (CW) harvested from a cluster; the average fresh fruit weight (CAW), determined as the total fresh weight (CW) divided by the total number of fruits harvested from the cluster (NFC); and the thermal time at which the first tomato of a particular cluster was harvested, expressed in growing degree days with a 10 °C base temperature (CGDD).

**Table 2.** Phenotypic descriptors evaluated in this study.

| Acronym | Description | Unit |
|---|---|---|
| | Vegetative | |
| SD | Stem diameter | mm |
| | Phenology and yield | |
| CGDD | Average growing degree days per cluster | growing degree days per cluster |
| FRU | Average number of harvested fruits per cluster | fruits per cluster |
| CLU | Total number of harvested clusters per plant | clusters per plant per $m^{-2}$ |
| NFC | Total number of fruits per cluster | fruits per cluster |
| NFM | Total number of fruits per plant | plant fruits per $m^{-2}$ |
| FFW | Average fresh fruit weight of all harvested fruits | grams per fruit |
| CW | Total fresh fruit weight per cluster | grams per cluster |
| CAW | Average fresh fruit weight | grams per fruit |
| Y | Fresh fruit yield | grams per $m^{-2}$ |
| | Flower | |
| NP | Number of petals | number |
| OL | Ovary length | mm |
| OD | Ovary diameter | mm |
| STL | Style length | mm |
| PL | Petal length | mm |
| AL | Anther length | mm |
| STE | Stigma exertion | mm |
| | Inflorescence | |
| NFL | Number of flowers in the inflorescence | number |
| NSF | Number of set flowers in the inflorescence | number |
| | Fruit | |
| NLC | Number of locules in a tomato of the third cluster | number |
| WG | Fresh weight of a tomato in the third cluster | grams |

### 2.5. Data Analysis

A principal components analysis, without yield, was performed using the R package "PCAmixdata", to establish the correlation circle between the traits evaluated and the yield components. Generalized linear mixed models were estimated for the variables.

Following Bolker's [31] guidelines, all variables were evaluated for a design of 3 blocks with subsampling. The blocking effect and the subsampling of each plot nested in the blocks were taken as random effects, while the accession was taken as a fixed effect. As indicated by Bolker [31], a distribution and a link function were selected: Gamma or Gaussian distributions for continuous variables, and Poisson for discrete variables. Residuals were graphically checked for normality. The models were estimated with the R package "lme4" [32] using the Laplace approximation, selecting the most positive likelihood

with normally-distributed residuals and homoscedasticity of variance. Analysis of the traits data from the genetic resources trial was made using the mixed linear model:

$$Y = X\beta + Zu + e \qquad (2)$$

where Y is the vector of observations, $\beta$ is the vector of fixed effects, u is the vector of random effects, X and Z are the associated design matrices, and e is a random residual vector. The assumed distributions of the random effects are e ~ MVN(0, R) and u ~ MVN(0, G), where MVN($\mu$, V) denotes the multivariate normal distribution with a mean vector $\mu$ and a variance-covariance matrix V. Blocks, plots nested in blocks, and genotypes were considered random effects. When genotypes are regarded as random effects, genotypic effects become part of u and are estimated by best linear unbiased prediction (BLUP). In multi-environment trials and single trials with balanced and unbalanced blocks, BLUP has been shown to outperform the estimation of genotypic effects as fixed effects estimated by best linear unbiased estimation (BLUE) [33]. Breeding values were estimated using BLUP, following Merk's [34] guidelines for tomato traits. The genetic correlations were then estimated, as with bread wheat yield, yield components and associated traits [35], wild radish flower traits [36], and switchgrass biomass yield traits [37], by correlating the traits with the breeding values calculated using BLUP. Spearman correlation analyses were performed using the Python package, Scipy. Spearman was selected in order to include extreme values, as a less sensitive correlation than Pearson, with regard to extreme values.

## 3. Results

### 3.1. Yield and Yield Components

The fruits of 365 plants were evaluated. Yield components showed great diversity between and within landraces, with a mean yield of 3297 g m$^{-2}$, showing high variation among the accessions (Table S2). The maximum measured yield was 9334 g m$^{-2}$ (SLY30), with no observed relationship between the year of collection and the yield of the plants. Distinctiveness of fruits among accessions was observed (Figure 1), with some landraces showing diverse fruit morphologies among clusters of the same plant (Figure 1B).

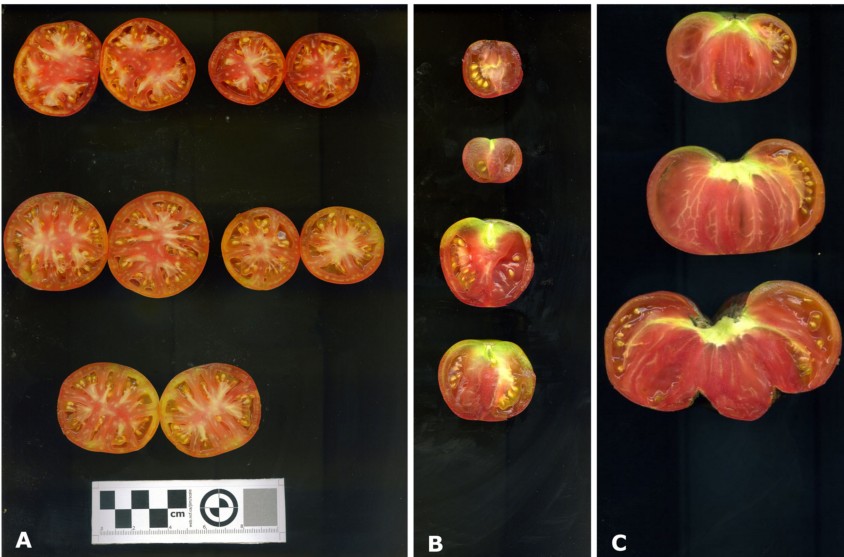

**Figure 1.** (**A**): Fruits from the third, fourth and fifth clusters, from top to bottom, harvested at one event from a single plant of accession SLY49. (**B**): Fruits from the third cluster of different plants of the accession SLY49. (**C**): Fruits from the third cluster from different plants of accession SLY157.

Yield-component correlations are shown in Table 3. Both yield components, the number of fruits (NFM) and the fresh fruit weight (FFW), had a significant positive phenotypic correlation with yield (Figure 2E,F), with NFM showing a higher phenotypic correlation

with yield (0.58***), and being the only yield component with a significant genetic correlation with yield (0.46*). A moderate and negative trade-off ($-0.66$***) between FFW and NFM was observed among genotypes. Assimilate partitioning for a given NFM was associated with the maximum FFW achievable (Figure 2A), following a diminishing-returns model, when associated with the yield achieved by the plants (Figure 2B,C), though a tendency in the number of clusters (CLU) explains part of the high variability observed in FFW and NFM (Figure 2E,F). The number of clusters (CLU) had a moderate genetic correlation with NFM (0.53**), with highly diverse architectures of clusters among plants. No genetic correlation was observed between CLU and fruits per cluster (FRU), with FRU showing a higher genetic correlation with NFM (0.80***).

**Table 3.** Yield and yield component Spearman correlations. *n* = 414.

| Correlation Coefficients | Y (Grams per m$^{-2}$) | | FFW (Grams per Fruit) | | NFM (Plant Fruits per m$^{-2}$) | | CLU (Clusters per Plant per m$^{-2}$) | | FRU (Fruits per Cluster) |
|---|---|---|---|---|---|---|---|---|---|
| Genetic (n = 24) | | | | | | | | | |
| FFW | 0.14 | ns | - | | | | | | |
| NFM | 0.46 | * | $-0.66$ | *** | - | | | | |
| CLU | 0.07 | ns | $-0.31$ | ns | 0.53 | ** | - | | |
| FRU | 0.28 | ns | $-0.69$ | *** | 0.80 | *** | 0.02 | ns | - |
| Phenotypic (n = 365) | | | | | | | | | |
| FFW | 0.30 | *** | - | | | | | | |
| NFM | 0.58 | *** | $-0.50$ | *** | - | | | | |
| CLU | 0.32 | *** | $-0.18$ | *** | 0.51 | *** | - | | |
| FRU | 0.36 | *** | $-0.45$ | *** | 0.69 | *** | $-0.19$ | *** | - |

Yield (Y), average fruit fresh weight (FFW), total number of fruits harvested per plant (NFM), number of clusters per plant (CLU), and average number of fruits harvested per cluster (FRU). Level of statistical significance: *: *p*-value $\leq$ 0.05; **: *p*-value $\leq$ 0.01; ***: *p*-value $\leq$ 0.001, ns: not significant.

### 3.2. Cluster Growth and Development

To evaluate the NFM components, a correlation analysis was made on a cluster basis. Of the 1678 clusters evaluated, a high degree of fruit polymorphism was observed between and within clusters of the same plant (Figure 1A), and within plants of the same landrace (Figure 1B). Clusters of the pink beefsteak tomatoes (i.e., SLY157) showed only double-flowers (fasciated or marigold-type flowers) (Figure 1C), whose fruits are characterized by a high number of locules, and are asymmetrical with two masses joined by a smaller central body. Ripening of higher clusters before the lower ones was common for all the plants evaluated, and associated with clusters with late fruit setting, clusters that yielded no fruits, and multiple clusters ripening together. The development pattern of each landrace (Figure 3) was very important for the on-field yield pattern characterization. Each landrace had a distinctive accumulated yield during the season, concentrating the yield to a given number of clusters. The landraces that presented the lowest growing degree days (GDD) to the first cluster, such as the landraces SLY147 and SLY121 (Figure 3), had most of their yield concentrated in the first cluster, while most of the landraces concentrated their yield around the third cluster. This is consistent with a prolonged period of development due to lower temperatures at the beginning of the season, associated with higher assimilate accumulation. A pattern relating GDD to the first harvested fruit and total cluster fresh weight (CW) diminishment was observed (Figure 4), associated with a decrease in fruit number per cluster (NFC) (Figure 4A) and average fruit fresh weight of the clusters (CAW) (Figure 4B). A possible explanation is that as the temperature rises during the season, the development period of each successive cluster is diminished, creating different CWs for a single NFC. As already observed on a plant basis in FFW against NFM (Figure 2A), assimilate partitioning compensation exists within clusters (Figure 4C). Increasing the NFC is associated with decreasing the maximum CAW achievable within the cluster. These findings suggest a two-level compensation, one at plant level (FFW against NFM) and the

second at cluster level (NFC against CAW). The NFM components CLU and FRU help to distinguish the two levels of compensation; on a plant basis, compensation would be associated with the plants' development rate as CLU, while the within-cluster compensation would be associated with the number of set flowers as FRU.

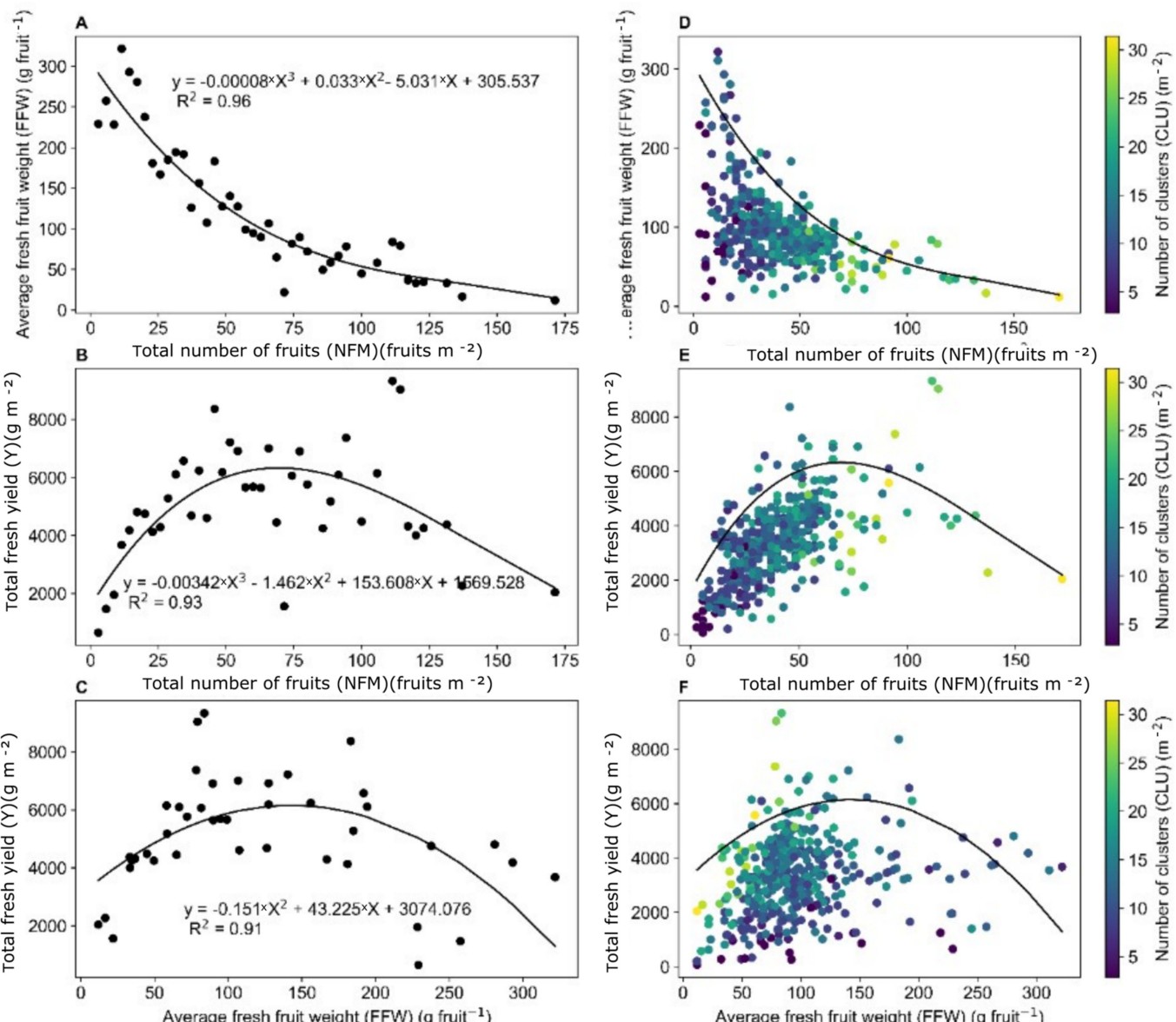

**Figure 2.** Tomato yield components. Left (**A–C**): Potential principal yield components; the models, fixed to the maximum values reached for each case, show a diminishing return for both yield components over yield. Right (**D–F**): Principal yield components of all the plants evaluated, colored by the cluster number per plant m$^{-2}$ (CLU), with the fixed curves.

### 3.3. Morphological and Phenological Traits

To further explore the morphological and phenotypical traits associated with the yield components, a series of correlations were made (Table S3). The traits were measured on a subset of 365 plants, however, all accessions and repetitions were represented. The traits evaluated had significant phenotypical correlations with at least three of the yield components (Figure 5), validating the importance of the descriptors. When phenotypic correlation coefficients were calculated between flower traits and yield, no significant correlations were found (Figure 5A). In contrast, flower traits showed significant correlations with yield components. The lengths of the anthers (AL), petals (PL), and ovaries (OL), as well as the

ovary diameters (OD), were phenotypically positively correlated with FFW, and negatively correlated with NFM and FRU (Figure 5B). Altogether, these traits would translate into larger flowers, likely associating with the within-cluster compensation as stronger sinks. The patterns of correlations seen for the inflorescence traits, NFL and NSF, were highly informative, being both significantly but weakly phenotypically correlated with yield. The number of flowers (NFL) was correlated to yield (Y) and all yield components, except for fresh fruit weight (FFW). The number of set flowers (NSF) had a strong correlation with all the components, including FFW, with higher correlation strengths than the correlations of NFL. The correlation between NFL and FFW was not significant, but the existence of a correlation between NSF and FFW suggests a late compensation between FFW and NFM. Thus, compensation would be associated with effective pollination followed by successful fruit setting. When observing the genetic correlations (Table S4), NFL correlated significantly only with the number of clusters (CLU), with a value of 0.41*, while NFS correlated significantly only with the total number of fruits (NFM), with a value of 0.51*, supporting the importance of the fruit-setting process as a yield driver.

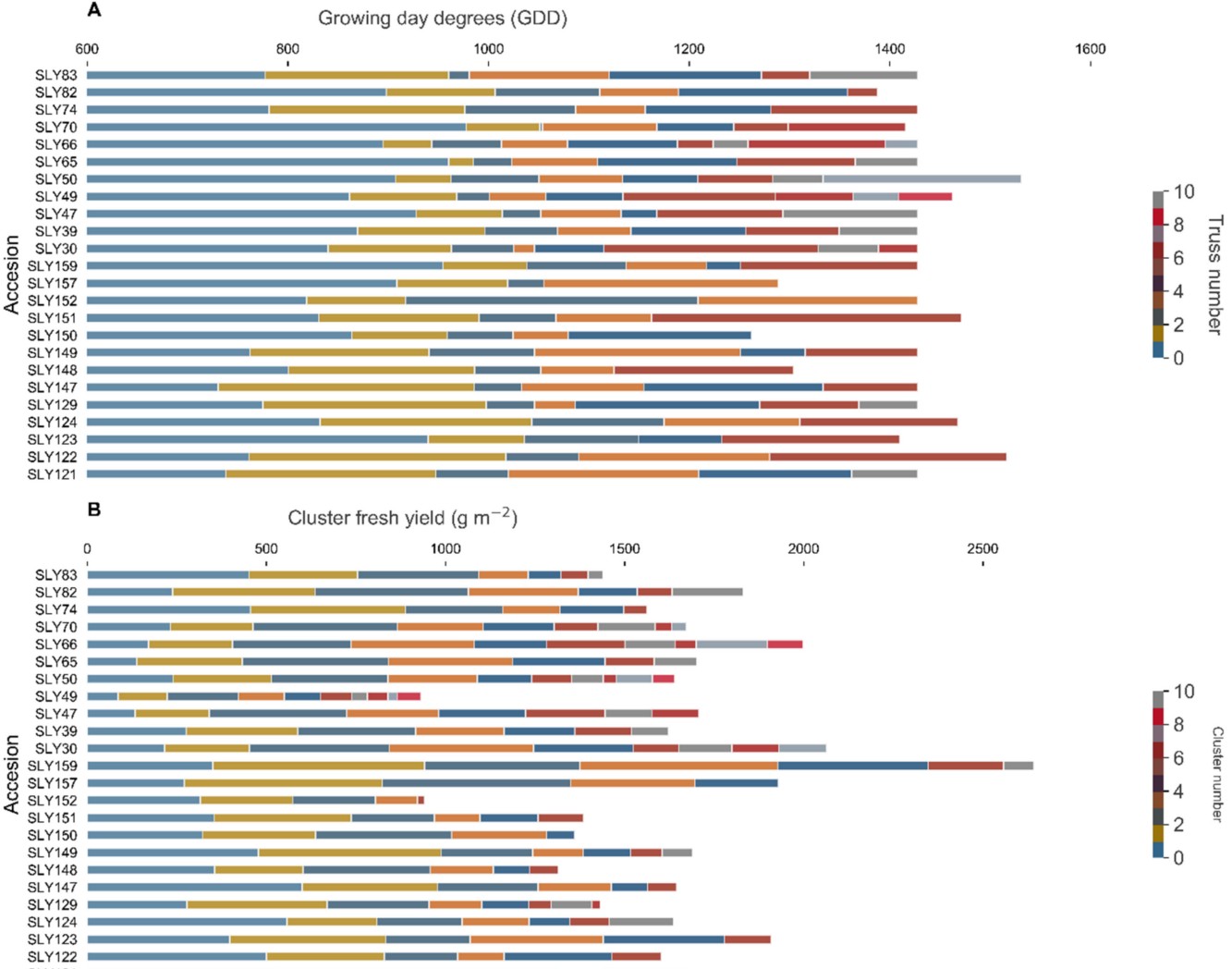

**Figure 3.** Yield patterns of tomato landraces grown in Santiago during the 2016–2017 season. (**A**): Accumulated growing degree days (GDD) between the harvest of each cluster, (**B**): Yield by cluster. The color in each bar represents the number of clusters. Each color bar is the average for each cluster per accession; color bars not seen are due to the population average being equal to, or lower than, the average of the consecutive cluster.

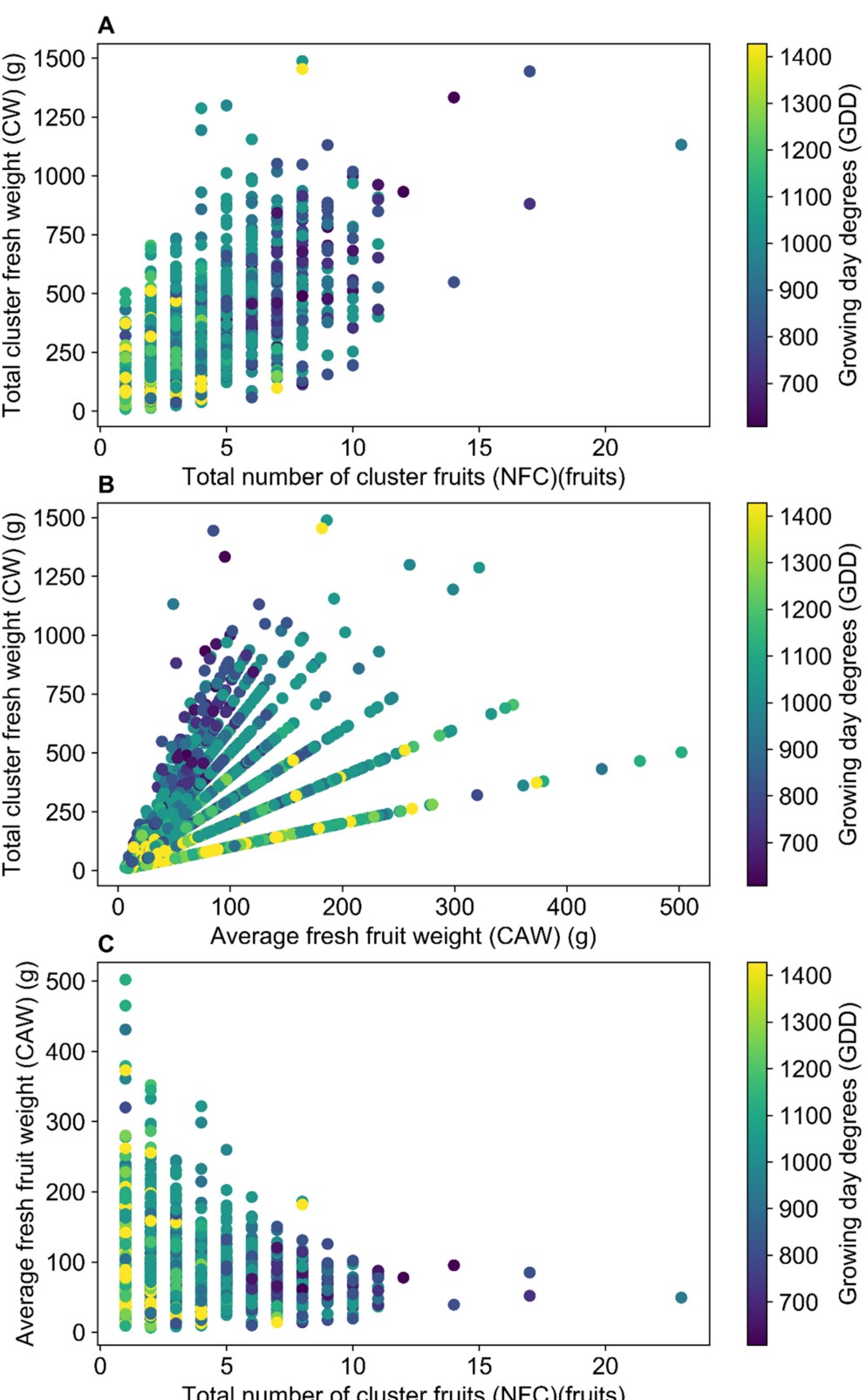

**Figure 4.** Cluster components to the first harvested fruit of each cluster of each plant individually. (**A–C**) Within cluster relationships between number of fruits, fruits fresh weight and clusters total weight, colored by growing degree days (GDD).

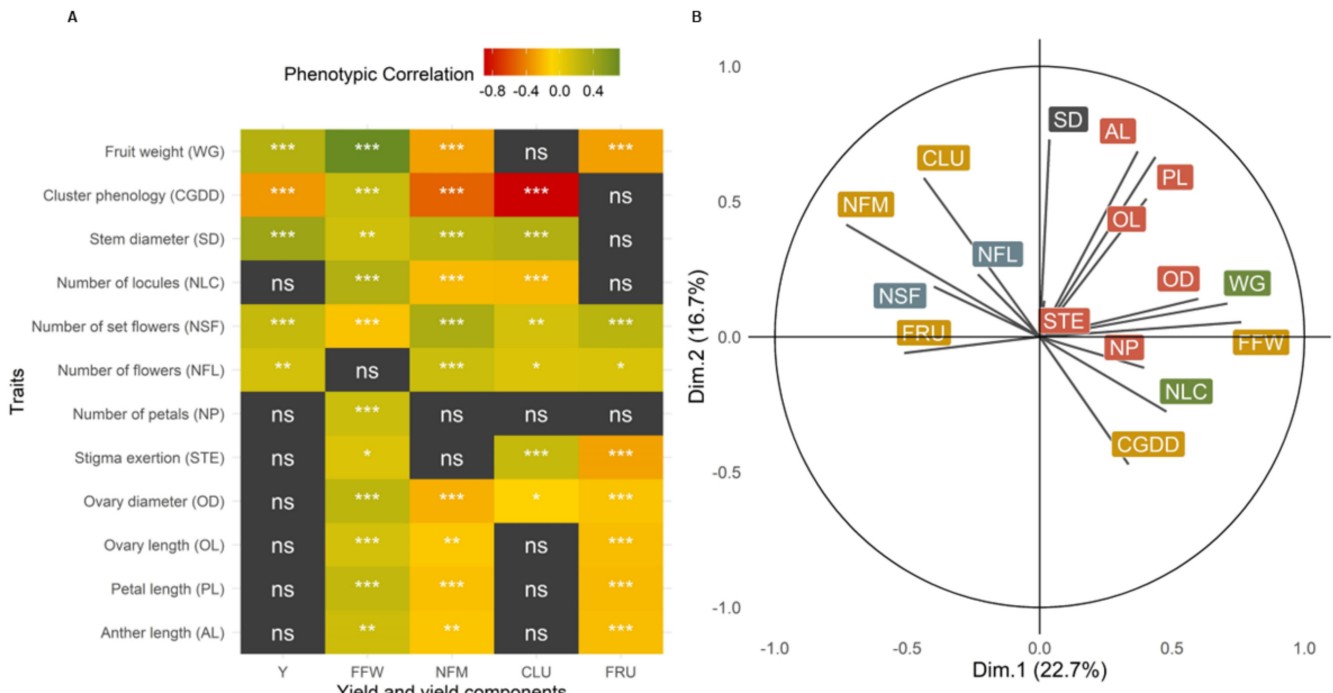

**Figure 5.** Correlation of reproductive traits and yield components. (**A**): Spearman correlation coefficient heatmap. Significance of the correlation *: *p*-value ≤ 0.05; **: *p*-value ≤ 0.01; ***: *p*-value ≤ 0.001, grey: not significant (ns); (**B**): Correlation circle of the principal components of the traits, yield not included. Colors differentiate traits of flowers (red), inflorescences (blue), fruits (green), and plants (yellow).

Flower characteristics had strong correlations among them (Table S3). Flower traits were phenotypically positively, but weakly, correlated with WG, which supports larger flowers having larger fruits. In addition, the number of locules (NLC) had the highest phenotypic correlation with WG (0.51***), an expected result as both traits were from the same fruits. However, when evaluated on a genotype basis only, PL (0.43*), NSF (−0.46*), and SD (0.45*) were significantly correlated with WG.

The fruit traits WG and NLC, evaluated on the third cluster, were significantly correlated with FFW (Table S4). Despite the observed lack of uniformity between clusters of the same plant, WG had a phenotypic correlation of 0.71*** with FFW, and was the only trait with a significant genetic correlation with FFW (0.79***). This supports the argument that the evaluation of the weight of a random tomato of the third cluster is a strong predictor of a plant's average fruit caliber. Moderate genetic correlations existed between WG and NFM (−0.54**), and WG and FRU (−0.62**), in contrast with NLC, which only significantly correlated with CLU (−0.45*) (Table S4). The growing degree days for a cluster to develop (CGDD) and the stem diameter (SD) had the highest correlations with fresh yield (Y), of −0.33*** and 0.43*** respectively. No trait had a genetic correlation with Y, although CGDD showed the highest genetic correlation with CLU (−0.90***) and a moderate genetic correlation with NFM (−0.52***). In contrast, SD only had a significant genetic correlation with FRU (−0.54**) (Table S4).

## 4. Discussion

### 4.1. Landrace Yields and Yield Components

The yields were within the parameters reported for tomato landraces. Yield and yield components are normally expressed per plant, but in order to compare between varieties, a correction by plant density is needed. Casals [14] reported yields and yield components for Spanish tomato landraces. Apparently small increases in plant values translate into high increments in yield and yield components, when plant density is considered. The

converted values of Casals [14] are within the yields obtained in this work, ranging from 6560 to 10,323 g m$^{-2}$. The yields achieved are also similar to those of the Dutch cultivars presented by Higashide [5], with yields ranging from 5000 to 9000 g m$^{-2}$. Consideration of the harvest period is also needed for comparison, as Casals [14] had a three-and-a-half-month harvest period, and Higashide [5] had three months for the whole experiment. In the present experiment, the harvest period was approximately four and a half months.

Fresh fruit weight (FFW) has been reported as a trait with high heritability in a broad sense, with estimated values of 0.87 [38] and 0.89 [39], appearing to be independent of plant density, although differential environmental effects and compensation profiles have been reported for yield and yield components of tomatoes [15,40]. Thus, although FFW is typically reported in large studies of genetic resources, there is no possibility of comparisons between different studies. The role of plant density in the determination of the NFM, and the compensation of FFW with respect to NFM, makes it necessary to report both values. The strong trade-off between NFM and FFW has been reported as a source-sink relationship [7], supporting the hypothesis relating the high FFW to limitations in the sink number (low number of fruits), maximizing the sink capacity of a single fruit per cluster. As further evidence, when a water deficit is applied after flowering, the fresh fruit weight increment has been connected to an increment in assimilate availability to each fruit, by a reduction in the number of flowers [23,41,42]. On the other side of the trade-off curve, a limitation of source (assimilates) when NFM is maximal, is compensated for with a low FFW. Fisher [26] demonstrated that increasing the cluster number generated a decrease in fruits per cluster (FRU) and fresh fruit weight (FFW), associated with competition for assimilates between the clusters due to the limitation in the source, while an increase in the assimilation rate with increasing fruit load (NFM), also shows that low sink strength can limit yield. Both source and sink strength can limit yield, giving further support to the source-sink origin of the trade-off observed as a diminishing-returns model (Figure 3). This result agrees with Fisher's [26] observations, with further evidence recently reported by Azevedo [24] in cherry tomatoes. The strong trade-off observed between FFW and NFM shows the importance of the distribution of the fruits. Equivalent NFMs present different compensations with FFW, regarding CLU and FRU (e.g., the compensation for ten fruits in four clusters is different from that of ten fruits in two clusters); the temporal distribution of fruits is an important component that might explain this difference.

The number of fruits per m$^2$ (NFM) is defined by multiple factors. The maximum NFM achievable is the NFL with a full set. Compensation with FFW seems to be significant only after fruit-setting (Figure 5). Further investigation should establish if NFM or NFL is the only coarse regulator of yield, following Slafer's [43] statements regarding grains per m$^2$ as the only coarse regulator of yield in wheat. On a genetic basis, tomato QTLs associated with yield components are promising. One is the fs8.1 QTL related to the NFL, supporting a genetic relationship between agromorphological traits (that can be evaluated early in the season) with yield components, as this QTL affects other traits as well, such as carpel length, fruit shape, yield, and harvest index [44]. In addition, the heterosis of the single flower cluster (SFT) allele also affects the NFL of tomato plants. Reported in relation to progress in yield improvement, there is a case where an agromorphological trait of a plant of low yield can be recombined in a high-yielding progeny [44]. Acevedo [45] proposed trait verification through the crossing of parents contrasting in four to five traits, is an important opportunity for tomato-yield improvement. The identification of accessions with contrasting yield components and agromorphological traits with high heritability could lead to novel improved varieties.

### 4.2. Plant and Cluster Compensation

Yield component trade-offs are associated with the timing of the processes involved in their determination [43]. On a genetic basis, the fw2.2 QTL has been described as the most important one affecting FFW, associated with a change in fruit cell number, as demonstrated by Bertin et al. [21], though it does not affect yield or harvest index. This is related

to compensation in fruit number on a plant basis (NFM), as the number of clusters (CLU) decreases with increasing FFW [7]. However, CLU is a phenological response, and as observed in the present work is a response associated with growing degree days (plant development). Growth and development are normally described as separated processes in plants, with mutually-regulated paths, associated with sugars [46] and light-sensing mechanisms such as cytochromes [47]. In the case of tomatoes, the hypothesis of assimilate compensation between FFW and CLU lacks a developmental perspective. This is shown in: (1) the compensation between clusters, as when a cluster is cut off, the weight of the adjacent clusters increases, without implying an overproduction of clusters [26]; (2) the yield compensation as CLU diminishes and FFW increases [7]; and (3) the stability of the cluster growing degree days requirement [4]. Increasing the development time of the clusters would increase the assimilate partitioning to each cluster, associated with higher calibers. Developmental issues have also been associated with production at low temperatures, where greater energy-efficient greenhouse production could be achieved with genotypes that can sustain their yields at suboptimal temperatures [1]. Thus, the study of the phenology of a diverse landrace is a promising field for yield potential improvement. Developmental differences among genotypes can help to generate a sustainable intensification of tomato growing.

Compensation of FFW with NFM components might be explained by the growth and development of fruits. Musseau [48] reported morphological tomato mutants with increased fruit weights that had the same locule numbers, raising questions about the limits of increasing FFW by increasing the locule number, and the effects on the phenological stages of the fruit of increasing FFW, though no correlation was observed between NLC and FFW among genotypes (Table S4). The negative correlations observed between flower elements and FRU could be associated with the higher sink strengths of larger flowers, as fruit cell number has been reported to be an early component of FFW [21]. Bertin [21] demonstrated that, with limited assimilates, the proximal fruits increase their sink strength to the detriment of the distal ones, followed by an increment in fruit cell number, generating a floral abortion or a delay in adjacent fruit development. However, no increment in yield was reported, similar to the compensation observed in the clusters among fruits. Overproduction of flowers, as an adaptive strategy, involves significant losses to the plant, therefore it is important that breeding strategies consider trait trade-offs [17,49].

*4.3. Development and Agronomic Management*

The tomato has been classified as a day-neutral plant, although many cultivars delay flowering under long-day conditions [50]. In this study, CGDD was used as a coarse estimator of plant-basis phenology, describing the average thermal time for the development of a cluster. This phenological stage would be a finer measure than days to ripening (8.1.2 descriptor; IPGRI 1996), as it uses the first ripened fruit on the last harvested cluster, instead of the first ripened fruit on the plant. A phenological state should be measured on a plot basis or cultivar basis, and that was considered for the development pattern (Figure 3), considering the average growing degree days to the first harvested tomato of each cluster of the accession. Multiple individual phenological processes occur during this development: (1) the formation of several vegetative metamers, typically up to 3–5; (2) inflorescence emergence; (3) the first flower of the opening inflorescence; (4) anthesis; and (5) fruit development to ripening. Van der Ploeg [1] used a similar approach, measuring the number of days between anthesis and harvesting of the first fruits per cluster, while Scholberg [13] measured the growing degree days per vegetative node and the number of vegetative nodes per inflorescence, although cluster number, as an NFM component, should consider only the number of clusters that yielded harvested fruits.

Tomato on-field cultivars have been associated with an increased number of flowers and early fruit set, creating smaller plants as the assimilates become limited early in the season, with an increase in the fruit set concentration for mechanized harvesting [6]. Breeding based on the lack of flower-repressing activity discovered in a self-pruning mutant

line a century ago is responsible for this burst of flowering, synchronized fruit ripening, and plant determination used for mechanical harvesting [50]. The concentration of yields at the beginning of the season ensures a great part of the yield (Figure 3). Barrios-Masias [6] described the increase in water-use efficiency (WUE), on a crop basis, indirectly associated with early flowering, smaller canopy size, and concentrated fruit set. In addition, early yield has a higher heritability than total yield, with heritabilities in the narrow sense of 0.73 and 0.06, respectively [39]. However, agronomic managements are also carried out on-field to decrease NFL. Floral pruning is a common practice that increases stem and lamina biomass, due to a change in the harvest index [20]. Further research is needed to determine the improvements associated with tomato farm yield, with respect to either genotype (breeding), agronomic management (plant density, floral pruning, greenhouse temperature), or genotype x agronomic management interaction.

**5. Conclusions**

Yield component decomposition is an important perspective for the comparison of studies and varieties. In tomato yields, a two-level compensation was observed. Development rate seems to be a main component of the compensation, on a plant basis, between the number of fruits (NFM) and fresh fruit weight (FFW), being correlated with most of the main yield components, whereas assimilate partitioning within clusters seems to relate more strongly to compensation within clusters, due to the number of fruits per cluster (FRU). Further multi-environment studies are needed to establish the plasticity of tomatoes.

Landraces showed great diversity of agromorphological and phenological traits, and this diversity might be key to yield improvement in tomatoes. The number of flowers (NFL), fruit setting (NFS), and growing degree days for a cluster to develop (CGDD) are the most promising traits for yield improvement. In a global climate change context, further study of trade-off relationships under abiotic stresses is important for identifying genotypes able to sustain their productivity.

**Supplementary Materials:** The following supporting information can be downloaded at: https://www.mdpi.com/article/10.3390/agronomy13020434/s1, Table S1: Soil and climate characteristics of the evaluation site, 2016–2017; Table S2: Yield, yield components and phenotypical traits Best Linear Unbiased Prediction (BLUP) for each accession; Table S3: Correlation among phenological and morphological traits; Table S4: Correlation among yield, yield components, and morphological traits.

**Author Contributions:** Conceptualization, A.D. and E.S.; methodology, A.D.; validation, A.D. and E.S.; formal analysis, A.D.; investigation, A.D.; resources, E.S.; data curation, A.D.; writing—original draft preparation, A.D.; writing—review and editing, E.S.; visualization, E.S.; supervision, E.S.; project administration, E.S.; funding acquisition, E.S. All authors have read and agreed to the published version of the manuscript.

**Funding:** This work was partially funded by the CONICYT scholarship N° 22170045 from the Programa Nacional de Becas de Postgrado—Magister 2017, the FONTAGRO Project (Code FTG/RF-15460-RG) and by the INIA/MINAGRI Conservation of Genetic Resources Program (Code 501453-70).

**Data Availability Statement:** Not applicable.

**Acknowledgments:** We would like to thank the La Platina Genetic Resources Unit staff for their assistance in developing this study.

**Conflicts of Interest:** The authors declare no conflict of interest.

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
