# Peer review of "Yield Components and Development in Indeterminate Tomato Landraces: An Agromorphological Approach to Promoting Their Utilization"

_agronomy, doi:10.3390/agronomy13020434_

Round 1

Reviewer 1 Report

agronomy-2130356

Comments for authors

The authors investigated the yield components and morphological properties of 24 indeterminated tomato landraces under field conditions. I could not find in the manuscript the evaluation of physiological properties that were indicated in the title. That is why I propose deleting the physiological traits in the title or evaluation of physiological properties related to e.g. photosynthesis and water consumption. A high degree of comparison with wheat is unfortunate it would be preferable to compare with varieties of tomatoes with determinate growth.

Details

In the abstract “parallel to cereal one” should be deleted whereas the investigation concern tomatoes.

Materials and methods section

page 5, 161-162 line: the formula for calculating CAW should be given here not in Table 2.

Table 2: delete the “in mm” in the description of flowering properties, since it is already in the unit column.

In Table 2 the formula of STE should be deleted that was already presented in the methodological chapter.

I suggest delete the Fig. 1 as

1) the data of the collection was shown from 1994 to 2010 and the varieties cannot be identified e.g.SLY30

2) the investigation was conducted from 2016 to 2017.

Fig. 6B: What do PG and RAC mean in the figure? Could there be a typo or other property?

page 13, 335 line: FRU might be CLU referring to Fig. 6B? It should be checked.

page 10, 262 line: SLY159 typo? Fig. 2 shows SLY149.

page 14, 361-372 lines: You should refer to Table S4 several times to make the results clearer.

373-378 lines: evaluation of Table S3 concerning on the relationship between flower properties and yield would be better to place after Fig. 6 and before Table S4.

Table S2: specify the unit of measurement of the yield

Table S4: specify the significance levels in the CLU column.

In the discussion chapter: the names of the authors should also be written before the reference number e.g. Fisher (25) reported…Similarly marked in rows 402,406,419,425,439,464,485,487,496.

406-409 lines: It is not clear whose result belongs to the (25) author or yours?

Author Response

Response to Reviewer 1 Comments

General Comments

Point 1. Moderate English changes required

Response 1: Text was reviewed and english was improved

Point 2. Introduction, references, methods, results and conclusions can/must be improved

Response 2. All sections were improved and suggestions accepted

Point 3. The authors investigated the yield components and morphological properties of 24 indeterminated tomato landraces under field conditions. I could not find in the manuscript the evaluation of physiological properties that were indicated in the title. That is why I propose deleting the physiological traits in the title or evaluation of physiological properties related to e.g. photosynthesis and water consumption.

Response 3. We agree with reviewer opinion. Title was changed to Yield Components and Development in Indeterminate Tomato Landraces: Agromorphological Approach to Promoting their Utilization

Point 4. A high degree of comparison with wheat is unfortunate it would be preferable to compare with varieties of tomatoes with determinate growth.

Response 4. Another comparison approach is provided.

Details

Point 5. In the abstract “parallel to cereal one” should be deleted whereas the investigation concern tomatoes.

Response 5. Abstract was improved and wheat reference was eliminated.

Materials and methods section

Point 6. page 5, 161-162 line: the formula for calculating CAW should be given here not in Table 2.

Response 6. Formula was  removed from Table 2.

Point 7. Table 2: delete the “in mm” in the description of flowering properties, since it is already in the unit column.

Response 7. Deleted

Point 8. In Table 2 the formula of STE should be deleted that was already presented in the methodological chapter.

Response 8. deleted

Point 9. I suggest delete the Fig. 1 as 1) the data of the collection was shown from 1994 to 2010 and the varieties cannot be identified e.g.SLY30  2) the investigation was conducted from 2016 to 2017.

Response 9. Deleted

Point 10. Fig. 6B: What do PG and RAC mean in the figure? Could there be a typo or other property?

Response 10. Abbreviation were corrected PG is WG and RAC is FRU

Point 11. page 13, 335 line: FRU might be CLU referring to Fig. 6B? It should be checked.

Response 11. Checked. FRU is correct, because in figure 6B was RAC.

Point 12. page 10, 262 line: SLY159 typo? Fig. 2 shows SLY149. Accession SLY157 and SLY159 are both beefsteak. Only SLY157 is shown in Fig 2C. Fig B2a shows accession SLY49. Line 262. Only refers to figure 2C.

Response 12. Explanation was improved in both, the text, and the figure.

Point 13. page 14, 361-372 lines: You should refer to Table S4 several times to make the results clearer.

Response 13. Suggestion was incorporated

Point 14. 373-378 lines: evaluation of Table S3 concerning on the relationship between flower properties and yield would be better to place after Fig. 6 and before Table S4.

Response 14. Paragraph was relocated as suggested

Point 15. Table S2: specify the unit of measurement of the yield

Response 15. Traits units are provided in material and methods, Table 2. Following the suggestion, unit were added to Table S2

Point 16. Table S4: specify the significance levels in the CLU column.

Response 16. done

Point 17. In the discussion chapter: the names of the authors should also be written before the reference number e.g. Fisher (25) reported…Similarly marked in rows 402,406,419,425,439,464,485,487,496.

Response 17. done

Point 18. 406-409 lines: It is not clear whose result belongs to the (25) author or yours?

Response 18. Is ours, the redaction was improved.

Reviewer 2 Report

The study ‘Yield components and development in…’ gives an insight on how plant morphological parameters are related with yield components. It provides new information that is worth publishing and relates with the tomato improvement. Below you may find some specific comments made.  

 The title refers to ‘..physiological approach’ but the manuscript describes a morphological approach

 Lines 1 2-14  are not clear

Line 15 also mention the number of genotypes studied

Line 15 … correlation with plant development..

Line 25. The term inderminate is mentioned for the 1st time. Why do you compare with dererminate what is in this case, there is missing information.

 Line 48 ‘… competition between source-sink balance’ perhaps more recent relevant studies may be cited. More than ever before, nowadays we need cvs not only producing more, but also  under stress conditions imposed by climate change. Here and also in the conclusion you could refer to the need in studying the trade-off between sinks imposed by stress factors affecting fruit development (ps in  Drogoudi PD & Ashmore MR. 14C assimilate allocation responses of fruiting and deblossomed strawberry to ozone. Being one stress factor,,,).

 Line 74 from tomato to cereals without explaining the reason and what we know about cereals

 Line 58: It is very confusing to talk about tomatoes and refer to grains…increasing 58 the number of spikelets spike-1, as a more neutral component [27] and, 2) crossing elite 59 parental lines contrasting in grain number and grain weight [28].

Degree day need to be replaced with Growing Degree Days (GDD)

 Fig 1 is not needed in my opinion.

 Suppl, Table 1. It is not menthioned what is the ‘La Platina Cental Valley’. Mean values of RH for every month should be placed next to temperature. It is not clear if there is a variation between the years studied and whether the years were typical in the studied area. It would have been better if the mean values of the previous 10 years in the area are also presented.

Suppl Table 2

Units need to be presented.

Fruit weight should be placed together with yield component

Suppl Table 3

What is WG? If fruit fresh weight this is stated as FFW in Supp Table 4 and WG in Suppl Table 2.

Author Response

Response to Reviewer 2 Comments

General Comments

Point 1. Extensive editing of English language and style required
Response 1: Text was reviewed and english was improved

Point 2. Introduction, references and results can be improved

Response 2. All sections were improved and suggestions accepted

Comments and suggestions

Point 3. The study ‘Yield components and development in…’ gives an insight on how plant morphological parameters are related with yield components. It provides new information that is worth publishing and relates with the tomato improvement. Below you may find some specific comments made.  The title refers to ‘..physiological approach’ but the manuscript describes a morphological approach

Response 3. Title was modified to "Yield Components and Development in Indeterminate Tomato Landraces: Agromorphological Approach to Promoting their Utilization"  

Point 4. Lines 1 2-14  are not clear

Response 4. Abstract was improved

Point 5. Line 15 also mention the number of genotypes studied

Response 5. Incorporated

Point 6. Line 15 … correlation with plant development.. ¿?

Response 6. Development was replaced with phenology.

Point 7. Line 25. The term inderminate is mentioned for the 1st time. Why do you compare with dererminate what is in this case, there is missing information.

Response 7. All comparisons with wheat were removed.

Point 8. Line 48 ‘… competition between source-sink balance’ perhaps more recent relevant studies may be cited. More than ever before, nowadays we need cvs not only producing more, but also  under stress conditions imposed by climate change. Here and also in the conclusion you could refer to the need in studying the trade-off between sinks imposed by stress factors affecting fruit development (ps in  Drogoudi PD & Ashmore MR. 14C assimilate allocation responses of fruiting and deblossomed strawberry to ozone. Being one stress factor,,,).

Response 8. Suggestion and reference were incorporated

Point 9. Line 74 from tomato to cereals without explaining the reason and what we know about cereals

Response 9. All comparisons with wheat were removed and introduction was improved.

Point 10. Line 58: It is very confusing to talk about tomatoes and refer to grains…increasing 58 the number of spikelets spike-1, as a more neutral component [27] and, 2) crossing elite 59 parental lines contrasting in grain number and grain weight [28].

Response 10. All comparisons with wheat were removed and introduction was improved.

Point 11. Degree day need to be replaced with Growing Degree Days (GDD)

Response 11. All terms were replaced in text and tables

Point 12. Fig 1 is not needed in my opinion.

Response 12. Fig 1 was removed

Point 13. Suppl, Table 1. It is not menthioned what is the ‘La Platina Cental Valley’. Mean values of RH for every month should be placed next to temperature. It is not clear if there is a variation between the years studied and whether the years were typical in the studied area. It would have been better if the mean values of the previous 10 years in the area are also presented.

Response 13. La Platina Central Valley was replaced by La Platina Experimental Station.

Monthly RH% data was incorporated

Point 14. Suppl Table 2. Units need to be presented.

 Response 14. Units are specified in Table 2 and were added to Table S2.

Point 15. Fruit weight should be placed together with yield component

 Response 14. IPGRI tomato descriptor consider fruit weight as a fruit descriptor. We prefered to leave it in fruit traits rather than yield traits to be consistent with that guideline.

Point 15. Suppl Table 3. What is WG? If fruit fresh weight this is stated as FFW in Supp Table 4 and WG in Suppl Table 2.

Response 15. FFW is Average fresh fruit weight of all harvested fruits. WG is Fresh weight of a tomato in the third cluster and is defined in Table 2. WG is correct in both, Table S2 and TableS4.